# Role of the Neuroendocrine System of Marine Bivalves in Their Response to Hypoxia

**DOI:** 10.3390/ijms24021202

**Published:** 2023-01-07

**Authors:** Elena Kotsyuba, Vyacheslav Dyachuk

**Affiliations:** A.V. Zhirmunsky National Scientific Center of Marine Biology, Far Eastern Branch, Russian Academy of Sciences, 690041 Vladivostok, Russia

**Keywords:** mollusks, bivalves, biogenic amines, stress, nitric oxide, hypoxia, hypoxia-inducible factor 1, neurotransmitters

## Abstract

Mollusks comprise one of the largest phylum of marine invertebrates. With their great diversity of species, various degrees of mobility, and specific behavioral strategies, they haveoccupied marine, freshwater, and terrestrial habitats and play key roles in many ecosystems. This success is explained by their exceptional ability to tolerate a wide range of environmental stresses, such as hypoxia. Most marine bivalvemollusksare exposed to frequent short-term variations in oxygen levels in their marine or estuarine habitats. This stressfactor has caused them to develop a wide variety of adaptive strategies during their evolution, enabling to mobilize rapidly a set of behavioral, physiological, biochemical, and molecular defenses that re-establishing oxygen homeostasis. The neuroendocrine system and its related signaling systems play crucial roles in the regulation of various physiological and behavioral processes in mollusks and, hence, can affect hypoxiatolerance. Little effort has been made to identify the neurotransmitters and genes involved in oxygen homeostasis regulation, and the molecular basis of the differences in the regulatory mechanisms of hypoxia resistance in hypoxia-tolerant and hypoxia-sensitive bivalve species. Here, we summarize current knowledge about the involvement of the neuroendocrine system in the hypoxia stress response, and the possible contributions of various signaling molecules to this process. We thusprovide a basis for understanding the molecular mechanisms underlying hypoxic stress in bivalves, also making comparisons with data from related studies on other species.

## 1. Introduction

Mollusks comprise the largest phylum of marine invertebrates. With their great diversity of species, various degrees of mobility, and specific behavioral strategies, they have occupied marine, freshwater, and terrestrial habitats, playing key roles in many ecosystems. Bivalves are common inhabitants of coastal marine waters. A number of marine bivalves (e.g., oysters, clams, and mussels) with wide geographic distribution are target aquaculture species, and are of high commercial value and scientific importance [1]. Bivalve aquaculture facilities are traditionally installed in coastal waters, where cultured animals can be exposed to permanent or periodic hypoxia, especially in the case of eutrophication expanding across coastal habitats, which leads to a decrease in their growth rates, impaired reproduction and development, diseases, and sometimes mass mortality [2,3,4,5,6]. Oxygen deficiency exerts the most adverse effect on juvenile mollusks by reducing, in particular, their growth, settlement, and survival rates [7]. In this regard, the knowledge of the molecular mechanisms associated with physiological and biochemical responses to hypoxia and resistance is essential to developing technologies for rearing certain mollusk species.

During their evolution, most marine bivalveshave been exposed to frequent short-term fluctuations in oxygen levels in their marine or estuarine habitats. This stress factor has caused them to develop certain adaptive strategies [8,9] to rapidly mobilize behavioral, physiological, biochemical, and molecular mechanisms that re-establish oxygen homeostasis [10,11,12,13,14,15].

Within the phylum Mollusca, due to the need for adaptive resistance to low dissolved oxygen levels, the behavioral and survival strategies against hypoxia havediverged widely across hypoxia-tolerant, slow-moving or sessile mollusk species that cannot use avoidance, and have to rely on physiological adjustments for coping with adverse conditions, in contrast to hypoxia-sensitive mollusks with enhanced mobility, which can avoid the hypoxic zone [2,16,17,18].

Hypoxia-tolerant intertidal species (e.g., mussels and oysters), the so-called oxygen conformers, reduce metabolic demand for O_2_ consumption as a response to environmental O_2_ levels [19,20,21], and also minimize energy demand [16,22,23,24]. These adaptations involve metabolic rate depression, the use of alternative glycolytic pathways that produce more ATP, the maintenance of high glycogen levels, and increases in the proton buffering capacities of tissues [12,24,25]. One of the behavioral responses exhibited by hypoxia-tolerant bivalves in the case of hypoxia is the closure of their shells and the regulation of the internal environment [26] (Figure 1).

Hypoxia-sensitive subtidal species (e.g., scallops), the so-called oxygen regulators [26], maintain O_2_ consumption independently of environmental O_2_ levels, up to the point where O_2_ consumption is limited to a level sufficient to maintain the aerobic process [23,27,28]. Therefore, they, like other mobile benthic animals, primarily exhibit developed behavioral and physiological mechanisms, such as the avoidance reaction, which allows for escaping the adverse effects of the hypoxic zone (Figure 1).

Though hypoxia-sensitive and hypoxia-tolerant animals show different homeostatic strategies to cope with oxygen deprivation [29], the major reaction of mollusks in response to environmental stressors, including hypoxia, is the neuroendocrine stress reaction [26]. This adaptive response, caused by stress-induced processes in the mollusk’s nervous system, induces behavioral and physiological changes in order to maintain homeostasis. The regulation of the stress response in mollusks involves the general mechanisms and signaling molecules preserved throughout evolution, having a molecular base similar to those in vertebrates [30,31,32]. As has recently been found, hemocytes in marine bivalves are an important component of the neuroendocrine-immune regulation that is undertakenin response to environmental stress [30]. Enzymes for catecholamine synthesis [33,34,35] and acetylcholine (ACh) degradation (acetylcholinesterase (AChE)) [36] are detected in molluskan hemocytes. The inhibitory effects of catecholamines on hemocytes’ functional responses have also been reported. Furthermore, hemocytes mediate the regulation of different effectors via specific receptors of neurotransmitters/hormones/neuropeptides/cytokines on the cell surface [30,37,38]. 

The neuroendocrine system in marine mollusks is sensitive to fluctuations in O_2_ concentrations. The synthesis of several neurotransmitters/modulators is regulated by O_2_-requiring rate-limiting enzymes. The hypoxia resulting from perturbations in the O_2_ homeostasis can affect neurotransmitter synthesis, thus causing altered neuronal functions and, consequently, affecting the physiological systems in mollusks and their stress response. 

Although the role of neurotransmitters in the regulation of functions in mollusks has long beena subject of research, their role in the hypoxia stress responses of marine mollusks is reported only in a few studies. In some marine invertebrates, variations in the level of biogenic amines (serotonin and dopamine) are considered as part of the hypoxia response, and act to arrest processes such as growth, reproduction, and immunity [39,40,41,42]. The changes of neurotransmitters lead to the redirection of bioenergetic resources to specific physiological functions (e.g., increased oxygen uptake, mobilization of energy substrates) that are immediately required for adaptation and survival under stress [43,44]. Several studies have reported the role of nitric oxide in invertebrates’ responses to stressful environmental conditions [45,46]. Other studies focus on gene expression during hypoxia responses and the role of a transcription factor referred to as the hypoxia-inducible factor-1 (HIF-1), which triggers and coordinates the up-regulation of multiple genes in response to low oxygen signals [47,48,49,50].

Even with recent advances, our understanding of the neuroendocrine system and its involvement in hypoxia stress responses of marine bivalves still remains very limited. Therefore, the study of the role of signaling systems in mollusks with different resistances to hypoxia is a key to the knowledge of the strategies for their survival. Understanding and determining this spectrum of neuroendocrine reactions to hypoxia will be useful for predicting the physiological condition of mollusks in aquaculture and their acclimation, which is important for the management of shellfish farms in coastal areas.

In this review, we have attempted to summarize data on some regulatory mechanisms of mollusks’ hypoxia resistance using available publications that consider the effect of hypoxia on the behavior and metabolic processes in mollusks, making also conceptual comparisons with data from related studies on other species. Here, we overview the current state of knowledge about the neuroendocrine regulation of oxygen homeostasis and the molecular mechanisms of hypoxia resistance on the basis of published data and the results of our research. The present review provides the current body of evidence elucidating the involvement of the neuroendocrine system in the hypoxic stress response, and the possible contributions of various signaling molecules in this process.

## 2. Biogenic Amines

Biogenic amines belong to the evolutionarily ancient signaling systems that are involved in the regulation of various physiological and behavioral processes, and in the processes of adaptation to various environmental factors in vertebrates and invertebrates exposed to stressful conditions [51]. These act not only as neurotransmitters and neuromodulators in nervous tissue, but also, depending on the situation, can be released into body fluids and act as neurohormones [51,52,53]. The major biogenic amines identified in mollusks are norepinephrine (NE), serotonin (5-HT), dopamine (DA), and epinephrine (E) [30,54]. In the neuroendocrine system of marine bivalves, the catecholamines and serotonin regulations play major roles in stress responses [55,56]. Being involved in the physiological response regulation in mollusks to maintain homeostasis through functional responses of the heart, hemolymph redistribution, and metabolic depression, these neuroendocrine messengers are immediately required for hypoxia tolerance.

### 2.1. Catecholamines

DA, NE, and E are present in various tissues of mollusks (in particular, bivalves), including the ganglion, hepatopancreas, and hemocytes [30,57,58,59,60]. The source of catecholamines in bivalves is predominantlythe neurosecretory neurons of the central nervous system (CNS) [30,61]; their secretion in hemocytes has also been reported [30,35]. Although DA, NE, and E have been identified in the mollusk CNS [30,54], DA isthe only catecholamine recorded inbivalve ganglia, according to immunohistochemical methods.

In the CNS, DA-ergic neurons have been found in *Mytilus edulis* [62], *Placopecten magellanicus* [59,63], and *Patinopecten yessoensis* [64,65]. A number of experimental studies have shown variations in the CA level in the CNS and hemolymph of mollusks exposed to various stress factors [61,66,67,68].

An increase in the CA concentration in the hemolymph is the primary adaptive neuroendocrine response exhibited by mollusks to any stress, which provides the metabolic and behavioral adaptation of these animals to adverse conditions [31,61,69]. In many invertebrates, including marine bivalves, hypoxia stimulates the secretion of the CA hormones [55,70] (Figure 2). In marine mollusks, NE and DA are released into the hemolymph within the first minutes after stress exposure [70]. The blood levels of both hormones in stressed octopus (*Eledone cirrhosa*) increased about 2–2.5-fold after 5 min of air exposure [70]. Concomitantly, a significant decrease in the number of circulating hemocytes was observed, whereas the hemocyte phagocytic activity and the superoxide anion production increased transiently between 5 and 60 min after the onset of the stress exposure. Scallops (*Chlamys farreri*) exposed to air for 12 h showed a significant increase in the hemolymph concentrations of E and NE [55]. During this period, the DA level significantly increased regardless of temperature (both at 5 and 17 °C) [55]. After entering the circulation, CAs contribute to reductions in the detrimental effects that are often associated with oxygen deficiency. 

In vertebrates and invertebrates, the beneficial effects of CAs are achieved, in part, by the modulation of the cardiovascular and respiratory systems [39,40,41,71]. The rise in CA levels initiates a series of compensatory physiological processes enhancing branchial O_2_ transfer and blood O_2_ transport.

In many bivalves, including *M. edulis* and *Crassostrea virginica*, the beating rates of lateral cilia are controlled by the branchial nerve via the reciprocal DA-ergic and serotonergic innervation originating from the cerebral and visceral ganglia [72,73]. DA is cilio-inhibitory and, with a few exceptions, it mostly decreases the ciliary beat frequency [72,74,75,76]. The activity of cilia can change in response to hypoxia, and is generally controlled by the nervous system [76]. The increase in DA level in the gills of *Cr. virginica* exposed to hypoxia had an inhibitory effect on gill ciliary beating [72]. Experimental evidence has shown that DA, applied directly to the ganglia or stimulating the branchial nerve (20 Hz, 2 ms duration, 10 V current), causes a terminal release of DA in the gill, decreasing the beating rates of lateral cell cilia [74,77,78]. In *Cr.virginica*, the direct application of dopamine to an isolated gill reduced the lateral ciliary activity in a dose-dependent manner (10^−7^ to 10^−3^ M), with 10^−5^ M being the ED_50_ dose [72]. However, as an exception, DA-ergic neurons induced an increase in the ciliary beat frequency during the hypoxia response in embryos of the snail *Lymnaea* [79]. In sea urchin embryos, DA increased the swimming speed, apparently through a cilioexcitatory effect [80,81]. In experimental long-time air exposure, the DA concentration was observed to decrease. 

In scallop (*Ch. farreri*), a significant decrease in the DA concentration in hemolymph was observed after 24 h of air exposure, which may be due to the moribund condition caused by the long-time air exposure that resulted in the disruption of DA responses [55]. A significant decrease in DA concentration was also observed in muscles of mussels (*Perna perna*) exposed to air for 24 h [82]. This effect is probably related to the contraction of adductor muscles for avoiding desiccation, as DA has been reported to cause acontractile effect in the adductor muscle of the freshwater mussel *Anodonta cygnea* [83]. 

Decreased tyrosine hydroxylase activity and dopamine deficiency are the major pathogenetic links in stress development in vertebrates and invertebrates [84]. Experiments using theneurotoxin 6-hydroxydopamine and the organic pesticide rotenone have shown the degeneration of a significant number of DA-synthesizing neurons and the dopamine deficiency in the CNS in insects [85] and mollusks [86], which caused the disturbance of their behavioral and locomotor reactions. 

### 2.2. Serotonin

Serotonin (5-hydroxytryptamine, 5-HT) is a neurotransmitter implicated in a wide range of physiological and behavioral processes in both invertebrates and vertebrates [87,88,89,90]. 5-HT is a major neuromodulator of motor behaviors in many species of invertebrate phyla, including mollusks [91,92,93,94]. It has been shown to modulate cognitive functions and play a fundamental role in the modulation of stress-induced excitability (arousal), in defensive behavior, in the modulation of aggressive behaviors, and in anxiety control [94,95]. 

5-HT and its receptors have been identified in the CNS in vertebrates and all groups of invertebrates, includingmollusks [62,94,96,97,98,99,100,101]. In the bivalve CNS, the organization of the 5-HT systems and 5-HT content differs between species and sexes, and is subject to seasonal variations [96,101,102,103]. Hypoxia exposure causes the 5-HT-immunoreactivity level to decrease in the ganglia and increase in the gills and other non-nervous tissues [72,104] (Figure 3). These data agree with the long-established fact that 5-HT has a cilio-excitatory and metabolic stimulatory effect on the gills of several bivalve mollusks [62,72,74,75,105,106] (Figure 2). 5-HT-immunoreactivity has been detected in ciliary nerves in most groups of ciliated animals [107], and similarities in the regulation of ciliary locomotion across different groups of animals have been shown [76]. In the gills of *Cr. virginica*, 5-HT activated movements of lateral cilia at a frequency proportional to the neuromodulator concentration, whereas DA had an inhibitory effect [72]. However, when the organism flushed the mantle cavity without feeding, the beating of laterofrontal cilia was arrested by high concentrations of 5-HT released from the serotonergic fibers [72].

Experimental studies have repeatedly confirmed the involvement of 5-HT in behavioral hypoxia adaptations that help mitigate the effects of hypoxia in encapsulated embryos of pond snails [79,108,109,110]. Encapsulated organisms are vulnerable to the adverse effects of hypoxia because of their inability to relocate through locomotion. In encapsulated embryos of *Helisoma trivolvis* and *Lymnaeastagnalis*, specific sensorimotor neurons release serotonin onto postsynaptic ciliary cells in response to hypoxia, resulting in faster ciliary beating and embryonic rotation [79,108,109] This induces more efficient oxygen diffusion due to increased stirring, and this maintains an adequate O_2_ supply during hypoxia [79]. The rotational behavior is a ventilation response that facilitates O_2_ diffusion to the embryo by reducing unstirred boundary layers [111]. This serotonin-mediated response acts through G-protein-coupled receptors. One of the receptors signals through the Gq pathway, leading to increases in intracellular Ca^2+^ [112]. The hypoxia response is also accompanied by increased cAMP levels in ciliated cells, mediated by another, Gs-coupled, serotonin receptor [113]. Similar cilio-excitatory effects were investigated in experiments on crustaceans, where 5-HT increased the rate of scaphognathite movement [114], which increased water circulation [114] and promoted a more rapid oxygen exchange in tissues as a response to higher oxygen demand.

Bivalves can obtain more oxygen by increasing the heart rate and dilating blood vessels in the case of hypoxia [115]. 5-HT is an excitatory agent that regulates the cardiac performance in various animals, including marine bivalves [116,117,118]. An increase in its level in response to declining oxygen leads to an increase in the heart rate and the amplitude of the heartbeat in mollusks [116]. The underlying mechanism for 5-HT’s effects involves the increase in cAMP [119]. In hypoxia-tolerant slow-moving or sessile mollusk species, a decrease in ambient *p*O_2_ is usually accompanied by a reduction in the heart rate and the amplitude of the heartbeat [116,120]. The heart rate in these mollusks decreases in response to a decrease in ambient *p*O_2_, presumably as an energy-saving response [120]. At intermediate levels of hypoxia, animals that regulate oxygen consumption may increase their heart rate. As has been shown by the Doppler ultrasonography method, the heart and respiratory rates in scallop (*Argopecten irradians*) increase when dissolved oxygen falls below 5 mg/L, which indicates that scallops rapidly adjust the circulatory rhythm to adapt to the stress [120]. Phasic changes in the heart rate in some species also appear to correlate with phasic movements of other organs. In scallops, moderate hypoxia (3 mg/L dissolved oxygen, DO) causes an increase in the blood flow, especially in the gill, to acquire more oxygen from the water and transport to other tissues; in *Perna viridis*, it causes an increase in the blood output to maintain the hemolymph circulation [121]. Neurotransmitters and mechanisms involved in the regulation of these changes have not been studied, however. The study of scallops by the Doppler ultrasonography method has shown that blood vessels dilate, and blood is redistributed to the gill for oxygen acquirement and to the adductor muscle for avoiding tissue damage [120]. In cases of severe hypoxia exposure in scallops, although the heart rate (HR) and blood velocity (PS) of all tissues largely increase, the blood flow volume (FV) in the tissue inevitably becomes reduced due to the constriction of the blood vessel, which means that the circulatory regulation has failed and functional damage is inevitable [120]. 

5-HT regulates the contraction and relaxation of the adductor and the anterior byssus retractor muscle (ABRM), and is involved in the regulation of complex changes in the protective behavioral reactions of mollusks during hypoxia. In bivalves such as the blue mussel *M. edulis*, smooth muscles such as the ABRM can be locked in the contracted state (i.e., “catch”), a crucial function that keeps the shell valves firmly closed during periods of air exposure [122,123]. This occurs following the initial activation of the muscle. This state is characterized by prolonged force maintenance in the face of low Ca^2+^, high instantaneous stiffness, a very slow cross-bridge cycling rate, and low ATP usage [124,125,126]. Tension is maintained until the serotonergic fibers release 5-HT, which stimulates the AC/cAMP/PKA system. Protein kinase A (PKA) is then responsible for the rapid muscle relaxation through the phosphorylation of twitchin, a myosin-binding protein [119,125]. The ABRM fibers in the catch state can be relaxed by serotonergic nerve stimulation or by the external application of 5-HT [123,125,127,128,129]. 5-HT induces an increase in the intracellular cyclic AMP (cAMP) concentration [130], which activates cAMP-dependent PKA to result in the phosphorylation of twitchin, a high-molecular-weight protein terminating the catch state [131].

Unlike slow-moving or sessile mytilids exposed to hypoxic conditions, scallops show an obvious escaping behavior, with the shells clapping frequently, which inducesa high demand for energy and oxygen [120]. However, the poor regulation ability of tissues under severe hypoxia means that the scallop may have lost this ability to escape the hypoxic “dead zone” to survive. 

In bivalves exposed to chronic hypoxia, the 5-HT level in the CNS has been found to decrease [104] (Figure 3), while the level of 5-HT in the hemolymph and mantle may increase significantly against air exposure stress [56]. In Pacific oysters (*Crassostrea gigas*) after exposure to air for 24 h, the high concentration of 5-HT in the hemolymph may decrease the apoptosis rate of hemocytes. The stimulation by 5-HT can enhance the resistance of oysters to oxidative stress under air exposure byincreasing the activity of superoxide dismutase (SOD), and reduce the accumulation of H_2_O_2_ in the hemolymph [56]. These protective effects of 5-HT were tested by estimating the survival rate of oysters after the stimulation of 5-HT in air, which showed anincrease in the survival rate of oysters upon exposure to air stress [56]. 

## 3. Acetylcholine

Acetylcholine (ACh) is one of the conserved neurotransmitters in the nervous system, and at the neuromuscular junction in vertebrates and invertebrates including mollusks [132,133]. ACh is synthesized in the cytosol from acetyl-coenzyme A and choline (produced via lipid metabolism) by the catalytic action of choline acetyl transferase (CHAT) [134]. Many indices of ACh neurotransmission recorded invertebrates, such as ACh content, CHAT activity, acetylcholinesterase activity, transporter mechanisms, and receptor-mediated responses, have also been detected in invertebrates [135,136]. 

Biochemical and histochemical studies on mollusks have demonstrated the presence of the enzyme synthesizing acetylcholine (CHAT) and the enzyme hydrolyzing it (acetylcholinesterase, AChE) in a multitude of taxonomic groups [36,104,132,135,136,137]. Furthermore, substantial homologies have been found between the ACh receptors cloned to date from invertebrate and vertebrate animals [138,139]. One AChE has also been identified in *Ch. farreri* [36], as have nicotinic ACh receptors (nAChR) in *Ch. farreri* [32], *Cr.gigas*, *Pinctada fucata martensii*, *Lottia gigantea*, *Aplysia californica*, *Octopus bimaculoides*, and *Helobdella robusta* [140], and a homolog of the muscarinic ACh receptor in *Cr. gigas* [32]. 

Cephalopods and gastropods with enhanced mobility have fewer nAChR genes than stationary bivalves [140]. The massive expansion and diversity of nAChR in stationary bivalve mollusks with simple nervous systems may be an adaptation to stationary life under a variable environment. In representatives of different mollusk groups such as cephalopods (*Octopus vulgaris*) [141]), including an octopus arm [142], pteropods (*Clione limacina*) [143,144], and bivalves (the scallop *Azumapecten farreri*) [104], ChAT-lir neurons were identified in the CNS, where most of them are localized in the motor centers and are involved in locomotor reactions [143,144], as well as in the escape behavior [141] (Figure 2). 

In the scallop *Az. farreri* exposed to hypoxia, an increase in CHAT in the motor neurons of the visceral ganglion [104], involved in adductor muscle contraction, correlates with changes in the adaptive behavior that manifestsas attempts to escape hypoxic water [145]. In stationary bivalve mollusks, ACh is also involved in the adaptive behavioral response to anoxia or low oxygen concentrations in seawater. It has been shown thatthe phosphorylation of PFK-1 alters the enzyme’s kinetic properties to convert it into a less active form in the anterior byssus retractor muscle (ABRM) of *M. edulis*, and to allow it to be mediated via cGMP [122]. An increase in cGMP occurs in the ABRM in response to ACh, which stimulates the contraction of this catch muscle [146]. The ABRM fibers can be madeto contract actively by cholinergic nerve stimulation or by the external application of ACh [147]. The catch state is established only after the removal of ACh, producing the maximum tension. This indicates that the development of ACh-induced tension to the maximum tension is the necessary prerequisite for the establishment of the catch state [148]. In bivalve mollusks, catch muscles such as the ABRM show an increased demand for energy during the first hours of recovery after valve closure, which is met by the activation of glycolysis [12,149].

Mollusks’ typical responses to hypoxia area variation in CHAT activity in all the ganglia and peripheral (branchial) nerves [104] (Figure 3). Cholinergic innervation has been characterized in ciliary bands of echinoderms, annelids, and mollusks [76,150]. In *M. edulis*, ACh is a modulator of the frontal cilia, with this effect being concentration-dependent [151,152]. In different groups of invertebrates, ACh decreases the ciliary beat frequency and increases closures [76]. The receptors to ACh involved in the ciliary movement in the gill plates have been found in scallop gills [36]. Due to the fact that bivalve mollusks’ gillshavethe highest exposure to the surrounding environment, the exceptionally high expression of nAChR genes in the gill may be an adaptation providing rapid response to dynamic environmental conditions [140]. 

Very little is known about the effect of chronic hypoxia on cholinergic transmission in the nervous system of invertebrates. Scallops (*Az. farreri*) exposed to long-term (12 h) anoxiashoweda significant increase in CHAT-lir in motor neurons of the visceral ganglion, as well as the involvement of neurons of the cerebral and pedal ganglia in the anoxia response (this did not show histochemical activity in the control) [104] (Figure 3). The mechanisms and functional consequences of the hypoxia-induced increase in the ACh synthesis enzyme in bivalves’ ganglia have not been elucidated. In recent studies, a significant increase in the production of CHAT has been recorded from the gills of the bivalve mollusks *Tapes decussatus* and *T. laeta* living in habitats with higher temperature and salinity and lower dissolved oxygen levels. The authors [153] assume thisto be an adaptive compensatory response for preventing the disruption of gill function. Previously, experiments on vertebrates showed that a 10 min complete ischemia (bilateral occlusion of the carotid artery in mice) caused choline (Ch) accumulation [154]. The major source of Ch accumulation during ischemia arises from the hydrolysis of Ch-containing phospholipids and phospholipid Ch-derived intermediates, with the contribution of ACh hydrolysis being small [155]. In the cholinergic neurons, the phospholipids containing Ch represent a large source of Ch that can potentially be used for ACh synthesis [156]. Neurons in mollusks contain large amounts of phospholipids [157]. Phospholipids such as phosphatidylethanolamine and phosphatidylcholine, exhibiting neurotrophic and neuroprotective effects, have been identified in mussels *M. edulis* [158]. Earlier, it was shown that Ch confers brain protection against ischemic stroke in mammals [159]. These data suggest that an increase in CHAT immunoreactivity in the scallop ganglia after long-termanoxia may be caused by an increase in the Ch levels, and is a conserved mechanism for protecting cells from hypoxia.

## 4. Nitric Oxide

Nitric oxide (NO) is an evolutionarily ancient, diffusible, gaseous low-molecular-weight signaling molecule occurring in all major groups of organisms, and a common regulator of metabolism [160,161,162,163] It is involved in many physiological functions, including cellular signaling in the nervous system, the regulation of vascular tone, responses to hypoxia, and nonspecific immune responses in both vertebrates and invertebrates [46,161,162,164,165,166]. NO is synthesized in cells from L-arginine and O_2_ by nitric oxide synthase (NOS), and acts as a nonspecific neurotransmitter and neuromodulator in the central and peripheral nervous systems. NO synthesis requires molecular oxygen and, therefore, its cellular production can be altered by hypoxia [133]. In mammals, NOS exists in three distinct isoforms: neuronal (nNOS, type I), endothelial (eNOS, type III), and inducible (iNOS, type II) [167]. 

NOS in mollusks shows a strong similarity with vertebrate nNOS in structure, with nNOS and iNOS in biochemical characteristics, and with iNOS in immunological features [168,169]. All the NOS forms require reduced nicotinamide adenine dinucleotide phosphate (NADPH) as an essential cofactor (an electron donor). Both NOS and NADPH-d are capable of transferring electrons from NADPH to tetrazolium salts and converting them into water-insoluble dark blue formazan crystals. The activity of NADPH-diaphorase (NADPH-d), a NADPH-dependent oxido-reductase, was shown to colocalize with nNOS immunostaining [170,171]. Positive NADPH-d staining has been successfully used as a marker for NOS in the CNS of both vertebrates [170,172] and invertebrates, including mollusks [173,174,175,176,177]. Furthermore, NOS-immunoreactivity has been recorded from different invertebrate phyla by the use of antibodies specific for the mammalian enzyme [162,176,178] (NADPH-d/NOS activity has been detected in the CNS, in peripheral tissues, and in hemocytes of marine invertebrates including mollusks [45,46,162,174,179,180,181,182,183,184] (Figure 2). In the hypoxia-tolerant mollusks *C. grayanus* and *Modioluskurilensis*, the NADPH-d/NOS activity in the CNS in control and hypoxia was significantly higher than in the hypoxia-sensitive species [104,177,181]. The increase in the NO level in invertebrate nerve cells under the effects of various environmental factors has been discussed in recent years [45,46,177,185]. The synthesis of NO is followed by its rapid diffusion within neurons and in adjacent cells [186]. Since the generation of NO occurs together with its release, a balanced NOS activity is a crucial step in the control of NO-mediated signaling [187]. Moreover, excessive NO release may lead to apoptosis, and exerts direct cytotoxic effects [186,187,188]. One of the ways to measure the detrimental effects of hypoxia in mollusks is to assess the severity of changes in the structure of nerve cells viaan ultrastructural examination. Higher morphological stability was found in NO-positive CNS neurons in hypoxia-tolerant mollusk species [189]. The hypoxia-tolerant bivalve species (oysters or clams) have a greater mitochondrial tolerance to hypoxia stress compared to hypoxia-sensitive species [24]. Mytilids belong to the group of marine invertebrates outstandingly tolerant to hypoxia and anoxia, endowed with specialized “anaerobic mitochondria” that can alternate between the use of oxygen (O_2_) and endogenous fumarate as the electron acceptor for anaerobic ATP production [46,190]. 

The survival strategy of these mollusks under hypoxia conditions is to reduce the total ATP uptake by switching to anaerobic metabolic pathways, or through metabolic depression [12,191,192,193]. NO plays a key role in reducing the metabolic rate in both vertebrates and invertebrates exposed to hypoxia [12,191,192,193]. To date, NO has been recognized as a potent mitochondrial regulator in vertebrate and invertebrate cells, where it reduces the oxygen affinity of cytochrome-*c*-oxidase (CytOx), the terminal electron acceptor of the mitochondrial electron transport chain [46,194,195]. NO binding to the enzyme is reversible and competitive with oxygen, and therefore depends on the cellular oxygen concentration [194,196]. Under nanomolar concentrations of oxygen in cells and reduced internal oxygen partial pressure (*p*O_2_) conditions, NO completely inhibits CytOx activity and, hence, mitochondrial and tissue respiration. It is also involved in switching neurons from aerobic respiration to glycolysis under conditions of reduced intracellular oxygen concentrations, a process that minimizes the production of reactive oxygen species [184,195].

NO also plays an important rolein improving the perfusion of hypoxic invertebrate tissues [46,184]. In mollusks, hypoxia increases the NO production in certain ganglia [104] (Figure 3) and in gill tissues [184]. The gills are the main organs of respiration in bivalves, where big hemolymphatic vessels run through the gill branches and filaments and connect the heart with the major tissues and organs. The local NO-dependent regulatory mechanisms that provide adequate blood flow in hemolymph vessels depending on the animal’s mode of life and habitat conditionshave been identified relatively recently in the gills of hypoxia-tolerant mollusks [46]. In mussels (*M. edulis*) inhabiting the intertidal zone under hypoxic conditions, NO is also generated in the muscle cells surrounding the hemolymph vessels of gill filaments. There, it functions as a hypoxic messenger and local vasodilator causing the blood vessels to dilate, which facilitates hemolymph flow and gas exchange at low *p*O_2_, and functionally stabilizes the rates of whole animal respiration [46]. In *M. edulis*, *p*O_2_-dependent NO generation is a key mechanism inwithstanding rapid environmental O_2_ fluctuations during low tide [46], when fast metabolic adjustments upon shell closure arerequired [184]. Unlike mussels, the infaunal clam *Arctica islandica* is a hypoxia-adapted species that actively regulates hemolymph and shell water *p*O_2_ at low levels (<5 kPa) through intermittent ventilation [197,198]. Obviously, there is no need for *Ar. islandica* to perform rapid adjustments of tissue oxygenation by NO-induced dilation of blood vessels. The NO formation itself remains constant under normoxia and hypoxia in the *Ar*. *islandica* gills. However, the active adjustment of mean internal *p*O_2_ to <5 kPa in these animals in vivo [197,198] appears to promote a stable NO concentrationin body fluids and tissues, and the lowering of mitochondrial respiration by NO-induced CytOx inhibition during self-induced burrowing and shell closure [184]. Thus, when the internal *p*O_2_ of tissues and hemolymph in *Ar. islandica* drops to values of ≤10 kPa during frequent burrowing periods or >24 h of shell-closure, accumulating NO may indeed diffuse from hemocytes into tissues and cells, reduce the oxygen binding at complex IV of the mitochondrial respiratory chain, and reducemetabolic rate [184]. Controlled metabolic shut down and a tiered reduction inelectron transport system (ETS) activities, including CytOx, may prevent significant reactive oxygen species’ (ROS) formation during hypoxic and anoxic transgression [46]. An ancient mechanism for controlling the respiratory electron transport under conditions of variable environmental oxygenation, typical of hypoxia-tolerant organisms inhabiting coastal marine environments (e.g., intertidal and subtidal habitats), has been developed [12,22]. In the hypoxia-sensitive, mobile scallop *Az. farreri*, the uNOS-lir level in the CNS increases only slightly with hypoxia. This contrasts with the NO-ergic activity in the ganglia [177], muscle cells, and hemolymph vessels of gill filaments in mytilids under both control and hypoxicconditions [46]. The significant differences in NO expression between the bivalve species may be related to their different modes of life and strategies of ecological adaptation [184].

In contrast to slow-moving mollusk species, scallops are less adapted to hypoxia owing to the high-energy cost of movement [120,199]. In the case of moderate hypoxia, scallops increase their respiratory and heart rates to maintain aerobic metabolism, which leads to an acceleration of hemolymph circulation [27,120]. However, in severe hypoxia, although the heart rate (HR) is largely increased, the blood flow volume (FV) in tissues drops, which causes the constriction of blood vessels. Thismeans that the circulatory regulation in these mollusks has failed, and functional damage becomes inevitable. It is likely that the scallops’ lower tolerance to hypoxia, as compared to that in hypoxia-tolerant mollusks, may be associated with a relatively low level of NO, and with the specific features of the organization and functioning of NO-dependent regulatory mechanisms that can affect the dynamics of blood flow. 

## 5. Hypoxia Inducible Factor-1α

The hypoxia inducible factor-1 (HIF-1) belongs to a family of highly conserved transcription factors that act as main regulators of oxygen homeostasis and theadaptive response to hypoxia [200,201,202,203]. HIF-1 regulates the expression of many genes involved in oxygen metabolism in response to hypoxic conditions [200,202,204,205,206]. HIF-1 consists of two subunits, α and β [200,201,202,207,208]. HIF-1β is constitutively expressed, without any effect of the oxygen level on its expression. 

The protein level of HIF-1α is highly regulated by oxygen tension [209]. The activity of HIF-1 is primarily determined by the expression of the subunit HIF-1α, but not that of HIF-1β. Under normoxia, HIF-1α is selectively degraded, while HIF-1β persists. During hypoxia, HIF-1α is stabilized, translocates to the nucleus, binds to HIF-1β, and initiates transcription [210,211,212], which triggers the expression of hypoxia-related genes [213] (Figure 4). 

The HIF-1-mediated system of oxygen-dependent signaling has also been identified in marine invertebrates [23,47,48,49,50,51,52,53,54,55,56,57,58,214,215,216], including several bivalve species such as *Crassostrea virginica* [215], *Cr. gigas* [216], *M. galloprovincialis* [50], the gastropods *Nassarius siquijorensis* and *N. conoidalis* [217], the small abalone *Haliotis diversicolor* [218], and the clam *Cyclina sinensis* [219]. The molecular characterization of HIF-1α partial coding sequences from various invertebrates (nematodes, oysters, and shrimp) and humans has shown a significant similarity of the sequences and the conserved key functional domains with the previously described isoforms from vertebrates and invertebrates. This also suggests the conserved critical role of these genes in the evolution of the oxygen-sensing pathway and homeostasis throughout the animal kingdom [50,220].

As in mammals, the HIF-1α of marine invertebrates is detected in multiple tissues, but its relative expression varies between different tissues [23,49,215,221]. The expression of the HIF-1α gene at both molecular transcription and protein levels indicates that various tissues within the same species may exhibit different hypoxic tolerances or oxygen demands, whereas the hypoxia intensity, as well as its duration, may affect HIF-1 in different ways [203,215,219].

The differences in HIF-1α transcript level after hypoxia exposure between various tissues in invertebrates may evincethe demand for a greater physiological response in certain tissues during adaptation to hypoxic conditions. In mobile crustaceans, the hypoxia exposure results in physiological or behavioral changes, such as an increased ventilation frequency and cardiac output [11,49,219,222]. Therefore, the marked upregulation of HIF-1α transcript levels in the heart of mantis shrimp (*Oratosquilla oratoria*) may reflect an HIF-induced enhancement of cardiovascular system functions, such as angiogenesis and vasodilation, to achieve efficient oxygen transport for providing survival under chronic hypoxia [49,223,224]. These results suggest that the upregulation of HIF-1α transcript levels in the two hypoxia-sensitive organs, the heart and the cerebral ganglion, is an important component of adaptation to chronic hypoxia in mantis shrimp and other marine invertebrates [49]. 

In many bivalves under normoxia, HIF-1α transcript levels were higher in the gills than in other tissues [215,219], most likely because the gill is a vital organ involved in oxygen detection and gas exchange [219,225]. Furthermore, the gills perform the important function of regulation in the progress of osmotic pressure adjustment, acid balance, and detoxification [226,227]. 

Under hypoxia, HIF-1α transcript levels areincreased in all tissues [29,228] and are particularly high in the gills of *Cr. virginica* [215], *H. diversicolor* [218], *Cy. sinensis* [219], *M. galloprovincialis* [50], *N. siquijorensis*, *N. conoidalis* [217] and *Ruditapes philippinarum* [14]. The transcript levels of HIF-1α mRNA in different tissues of mollusks significantly differedin time during hypoxia [219]. Thus, in the small abalone *H. diversicolor* exposed to hypoxia (2.0 mg/L DO at 25 °C) stress, the HIF-1α expression was upregulated in gills at 4, 24, and 96 h, and in hemocytes at 24 and 96 h [218]. In *Cy. sinensis*, the transcript level declined continuously after 12 h hypoxia [219].

The nervous system is an important component of the organism that requires oxygen. In mammals, HIF-1α plays a crucial role in protecting neurons from hypoxic/ischemic stroke. In the invertebrate nervous system, the function of HIF-1α is poorly understood [49]. In hypoxia-sensitive scallops (*Mizuhopecten yessoensis*) exposed to hypoxia, HIF-1α expression appears primarily in the nuclei of neurons of the cerebral ganglia [229]. The high sensitivity of these bivalve ganglia to hypoxia has been confirmed experimentally [230], with their involvement in respiratory metabolism also shown [231,232]. After 4 h of anoxia, the number of HIF-1α immunopositive neurons in the visceral ganglion sharply increases. The ganglion is an integrative center in mollusks that is involved in the respiration regulation, controls motor behavior, and plays a major role in metabolic processes and in the escape behavior under extreme conditions [233]. The expression of the HIF-1α factor in the cerebral and visceral ganglia controlling the critical functions in scallops is probably associated with the involvement of this factor in the adaptation of neurons to hypoxia. 

An increase in the HIF-1α content in the mammalian brain correlates with neuroprotective reactions, and prevents or reduces damage in moderate hypoxia/ischemia [234,235,236]. In the case of chronic and severe hypoxia, the expression of HIF-1α in the rat brain is reduced due to an increase in the rate of its degradation, which correlates with a decrease in the function of the mitochondria and apoptosis of neurons [237]. An increase in the HIF-1α expressionin neurons’ nuclei causes the activation of the genetic apparatus and primarily genes, which triggers a cascade of neuroprotective mechanisms that protect neurons, macro- and microglia, and the endothelium of brain vessels from damage caused by oxygen starvation [234,235,236]. In the mammalian brain, these mechanisms induce rapid and adequate responses to hypoxia through stimulation of the respiratory and vasomotor centers, and lead to the induction of genes necessary to provide the energy metabolism of cells [238,239].

The role of HIF-1α in the CNS of marine invertebrates in the formation of hypoxia tolerance has not been extensively studied. Currently, there are data on the HIF-1α expression in the brain of mantis shrimp (*O. oratoria*) during adaptation to chronic hypoxia caused by the anthropogenic pollution of the habitat [49]. Furthermore, HIF-1α-immunoreactive neurons that are involved in behavioral and metabolic reactions to hypoxia have been identified in the CNS of nematodes, *Caenorhabditis elegans* [240]. In scallops, anoxia has a pronounced effect on the activity of HIF-1, significantly increasing the expression of its regulatory oxygen-sensitive subunit HIF-1α in the ganglia neurons that control critical functions of the organism, which can provide the development of compensatory processes in hypoxia. This is confirmed by the results of a study on the metabolism and activity of oxidant enzymes in scallops [27,241]. 

## 6. Conclusions

The issue of adaptation to oxygen deficiency and its role in diseases has been studied for many decades using models of animals with different resistances to hypoxia. Despite the recent advance in invertebrate neuroendocrinology, very little is known about the neurohormonal regulation of this process in marine bivalves. Marine mollusks’ tolerance to hypoxia is provided by the integration of various signaling systems whose activation causes changes in the expression of neurotransmitters such as DA, 5-HT, CHAT, and NO. Their activity varies significantly between species living in different conditions and having different survival strategies, which indicates the different roles that they play in mollusks with different tolerances to hypoxia. Of particular interest are the dynamics of activity of 5-HT, uNOS, and the HIF-1α transcription factor in the ganglia, branchial nerves, and gills, which probably reflect their key roles in the regulation of gas exchange and cardioregulation in marine mollusks exposed to hypoxia. Data on the topography and dynamics of CHAT activity in hypoxia-sensitive scallops indicate a possible neuroprotective role of choline, which may be one of the mechanisms responsible for protecting nerve cells from hypoxia in mollusks. However, further studies are required to obtain physiological evidence of the involvement of DA, 5-HT, CHAT, NO, and HIF-1α in providing hypoxia tolerance.

## Figures and Tables

**Figure 1 ijms-24-01202-f001:**
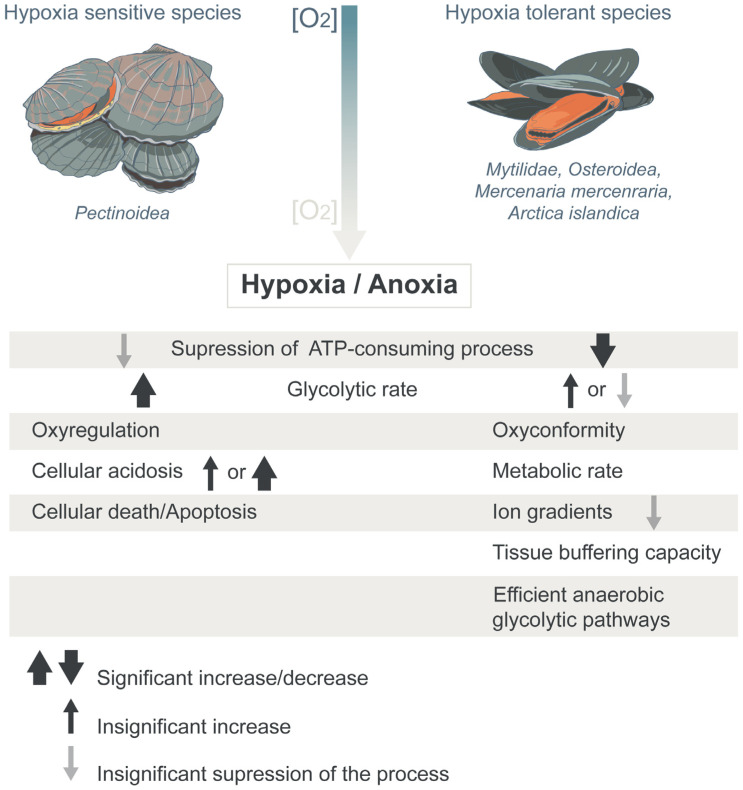
Division of marine bivalves on the basis of their metabolic response to hypoxia and the differences in the molecular mechanisms of their tolerance to low-oxygen conditions.

**Figure 2 ijms-24-01202-f002:**
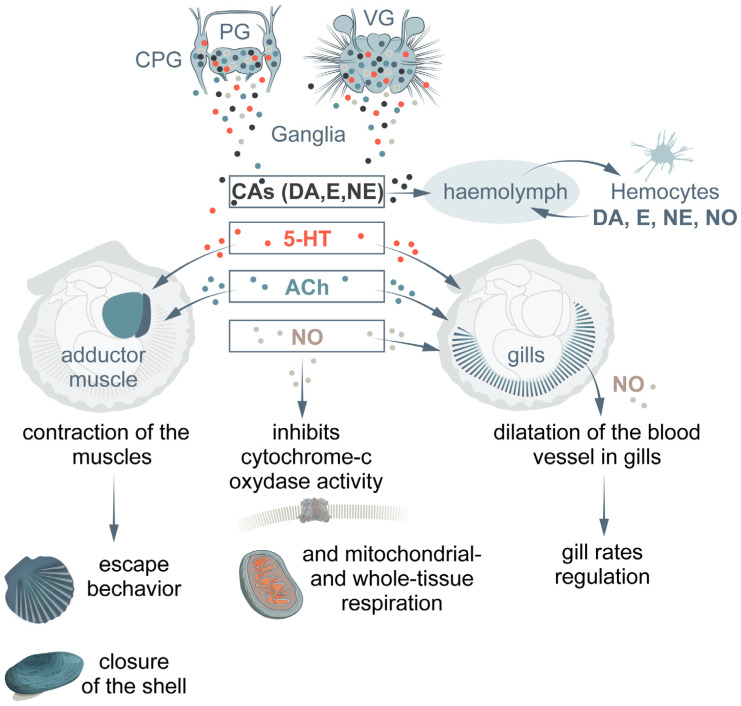
Diagram summarizing available data on the hypoxia stress control and the role of neuroendocrine regulation in the hypoxic response in bivalves. The involvement of CAs, 5-HT, ACh, and NO under hypoxic conditions is shown.

**Figure 3 ijms-24-01202-f003:**
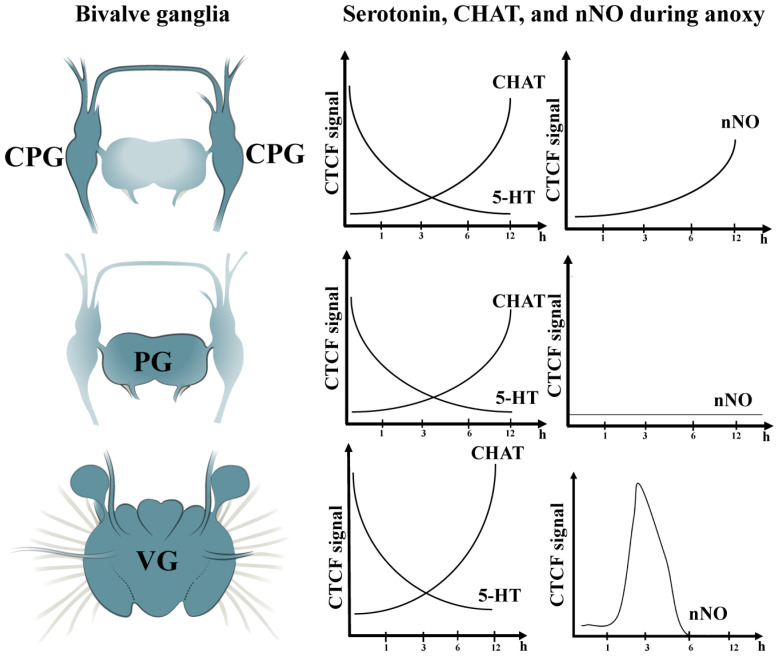
Temporary changes indistribution of neurotransmitters in scallop ganglia during air exposure-induced hypoxia. (**Left**) Quantitative representation of neurotransmitters. (**Right**) Variations in5-HT, CHAT, and uNOS in cerebropleural ganglia (CPG), pedal ganglia (PG), and visceral ganglia (VG) during hypoxia exposure (at 0 (normoxia), 1, 3, 6, and 12 h).

**Figure 4 ijms-24-01202-f004:**
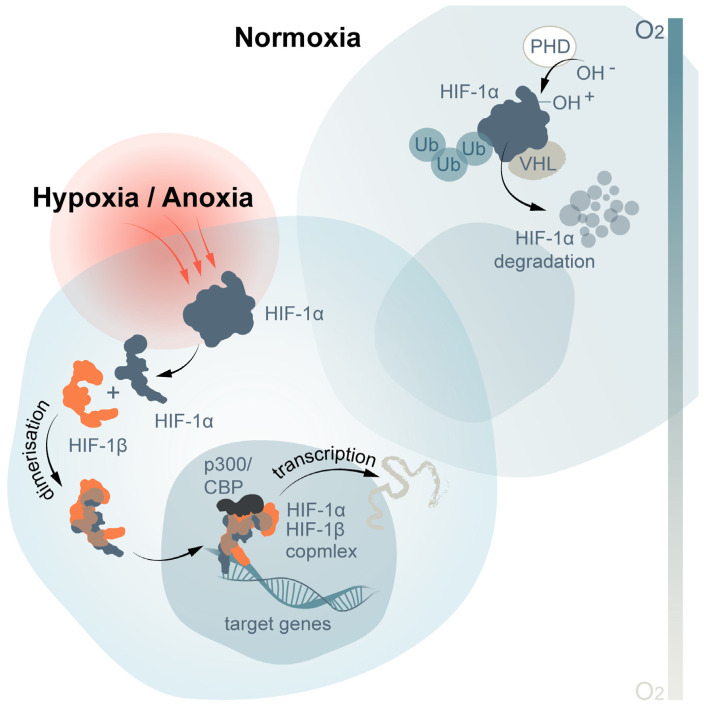
Involvement of HIF-1 in changes of gene expression under hypoxic conditions in bivalves.

## Data Availability

Not applicable.

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
