# Peer review of "Role of the Neuroendocrine System of Marine Bivalves in Their Response to Hypoxia"

_ijms, 2023, doi:10.3390/ijms24021202_

Round 1

Reviewer 1 Report

This manuscript proposes an extensive review on role of the neuroendocrine system of marine bivalves in response to hypoxia, the neuroendocrine system and its related signaling systems play crucial roles in the regulation of various physiological and behavioral processes in mollusks and can affect hypoxia tolerance. The neuro-endocrine stress response to hypoxia and the neuro-endocrine regulation of oxygen homeostasis have been summarized based on the current knowledge. The findings in the cited materials show that the marine mollusks’ tolerance to hypoxia is provided by the integration of various signaling systems and their activation causes changes in the expression of neurotransmitters such as DA, 5-HT, CHAT, and NO. As such, the matter is of interest, but this article still has some shortcomings as follows.

1. The references should not be appeared in the abstract.

2. Table 1 cannot be found in the manuscript.

3. In line 177-179, no references are cited.

4. In Figure 3, Ca (DA, E, NE) should be CAs (DA, E, and NE)

5. In line 229-231, there are only references for vertebrates

(Bacqué-Cazenave et al., 2020), and no references for invertebrates are cited.

6. The statement in the sentence is inconsistent with the figure in regarding the level of 5-HT, as can be seen from the figure, the level of 5-HT is reduced, but in the sentence, But the description is increased in the sentence in line 244

7.Many species that are not marine bivalves are mentioned,for example in line 613, 616, there are a lot of references that have been done on Marine bivalves, therefore, it is recommended to find references related to marine bivalves.

8. “Copmlex” should be complex in Figure 5

Author Response

Comments and Suggestions for Authors

This manuscript proposes an extensive review on role of the neuroendocrine system of marine bivalves in response to hypoxia, the neuroendocrine system and its related signaling systems play crucial roles in the regulation of various physiological and behavioral processes in mollusks and can affect hypoxia tolerance. The neuro-endocrine stress response to hypoxia and the neuro-endocrine regulation of oxygen homeostasis have been summarized based on the current knowledge. The findings in the cited materials show that the marine mollusks’ tolerance to hypoxia is provided by the integration of various signaling systems and their activation causes changes in the expression of neurotransmitters such as DA, 5-HT, CHAT, and NO. As such, the matter is of interest, but this article still has some shortcomings as follows.

Reply

Dear reviewer. We express our great gratitude to you for review of the MS and careful work on MS. We have reviewed all your valuable comments and made appropriate corrections in the text and figures. We hope that we have satisfied all your comments.

Best wishes

Vyacheslav Dyachuk

Elena Kotsyuba

1. The references should not be appeared in the abstract.

Reply

Sure, I have deleted all eferences in the abstract of the MS

  1. Table 1 cannot be found in the manuscript.

Reply

Sorry, this is our mistake. The text of the MS does not contain any Tables

  1. In line 177-179, no references are cited.

Reply

Malham et al., 2002; Chen et al., 2008 are cited in MS

  1. In Figure 3, Ca (DA, E, NE) should be CAs (DA, E, and NE)

Reply

We have changed

  1. In line 229-231, there are only references for vertebrates

(Bacqué-Cazenave et al., 2020), and no references for invertebrates are cited.

Reply

Agree! we have added some Refs for invertebrates

  1. The statement in the sentence is inconsistent with the figure in regarding the level of 5-HT, as can be seen from the figure, the level of 5-HT is reduced, but in the sentence, But the description is increased in the sentence in line 244

Reply

Right! Hypoxia exposure causes the 5-HT-immunoreactivity level to decrease in the ganglia and increase in gills and other not nerve tissues. We have changed.

7.Many species that are not marine bivalves are mentioned,for example in line 613, 616, there are a lot of references that have been done on Marine bivalves, therefore, it is recommended to find references related to marine bivalves.

Reply

We agree. We have added two paragraphs (line 613-622)

  1. “Copmlex” should be complex in Figure 5

Reply

Done

Thanks for the revision

Best wishes

Dr Vyacheslav Dyachuk

Dr Elena Kotsyuba

Reviewer 2 Report

A thorough review, well done.  The manuscript needs only minor copy editing to improve clarity and readability.  A copy of the PDF with my editorial suggests is attached. 

Author Response

Dear Reviewer!

We want to express huge thanks to the reviewer who did a great job to improve this manuscript. Thank you for your time and a detailed correction of MS.

WE have taken into account all your comments and corrected the text.

Best

Vyacheslav Dyachuk

Ekena Kotsyuba

Round 2

Reviewer 1 Report

no more comments.